# Processing Real-Life Recordings of Facial Expressions of Polish Sign Language Using Action Units

**DOI:** 10.3390/e25010120

**Published:** 2023-01-06

**Authors:** Anna Irasiak, Jan Kozak, Adam Piasecki, Tomasz Stęclik

**Affiliations:** 1Deartament of Pedagogy, Jan Dlugosz University in Czestochowa, Al. Armii Krajowej 13/15, 42-200 Czestochowa, Poland; 2Department of Machine Learning, University of Economics in Katowice, 1 Maja 50, 40-287 Katowice, Poland; 3Łukasiewicz Research Network—Institute of Innovative Technologies EMAG, Leopolda 31, 40-189 Katowice, Poland

**Keywords:** action units, automatic translation, sign language, entropy of real data

## Abstract

Automatic translation between the national language and sign language is a complex process similar to translation between two different foreign languages. A very important aspect is the precision of not only manual gestures but also facial expressions, which are extremely important in the overall context of a sentence. In this article, we present the problem of including facial expressions in the automation of Polish-to-Polish Sign Language (PJM) translation—this is part of an ongoing project related to a comprehensive solution allowing for the animation of manual gestures, body movements and facial expressions. Our approach explores the possibility of using action unit (AU) recognition in the automatic annotation of recordings, which in the subsequent steps will be used to train machine learning models. This paper aims to evaluate entropy in real-life translation recordings and analyze the data associated with the detected action units. Our approach has been subjected to evaluation by experts related to Polish Sign Language, and the results obtained allow for the development of further work related to automatic translation into Polish Sign Language.

## 1. Introduction

Legislation currently in force in the European Union and around the world requires that people with disabilities be treated equally and be provided with unrestricted communication and access to information [1]. In the Polish context, two documents have been enacted over the past few years that specifically regulate information and communication accessibility, and digital accessibility for people with disabilities, including Deaf people [2,3]. The documents mentioned above provided the impulse to search for technological solutions to remove or minimize existing barriers in this area.

It is worth mentioning that, for many Deaf people, a sign language is the one first and dominant in everyday life while the national language is the second one and the level of personal proficiency in it varies [4]. In addition, the lack of a universally-valid written form of a sign language promotes the use of digital technologies and, at the same time, overcomes the barrier of this lack. It allows Deaf people to communicate at a distance that just a few decades ago was impossible or greatly hindered. A number of studies about a sign language recognition, generation and translation are currently underway. Their purpose is to help break down barriers for a sign language users in everyday life. In this regard, the topics of an ongoing research generally concern the translation of a national language into a sign language as input using text, sound, or image [5,6]. There is also emerging research into reverse translation, an example of which is a solution described in [7]; if that is applied the system can recognise sign language poses and translate through avatars in the form of talking faces. A lot of work is also focused on developing bidirectional communication capabilities by creating solutions that translate spoken languages into sign languages and can also recognise sign languages as in the works of [8,9,10]. In addition, in the aspect of Polish Sign Language, research work has been conducted to facilitate the communication of a Deaf person who communicates using a sign language with a person who does not know such a way of conversation [11,12,13,14,15].

Many of the works indicate the great importance of non-manual components in sign communication. This language does not rely only on manual gestures but also on facial expressions and other non-manual markers. It poses a major challenge to researchers working on the topic of sign language analysis and synthesis. The examples of proposed solutions for dealing with existing difficulties in designing facial expression recognition systems are provided by research into Japanese and Brazilian Sign Language [16,17].

In addition, many research projects are being conducted in the area of sign language synthesis and developing solutions that practically use developed systems, especially for the development of signing avatars, which is also the subject of a project we are currently conducting. In [18], the author indicates three interrelated threads related to the best way to portray the linguistic and paralinguistic information expressed on a signer’s face. First, it is a linguistic approach to facial expressions, and because of including that an avatar must be required to communicate intelligibly; second, computer graphics, which should provide the right tools and technologies. A third theme addresses the topic of sign language representation systems from the point of view of their ability to represent non-manual signs and facial expressions. Non-manual signs, facial expressions, and the generation of synthetic emotions have also been addressed in papers [19,20,21,22,23]. Those articles also describe efforts to improve the quality, realism, and facial expression in sign language animation.

In Poland, additional potential for research in the above-mentioned areas is provided by extensive corpus-based research into Polish Sign Language conducted at the University of Warsaw by the Section for Sign Linguistics [24]. In the project of corpus research, for a period of 10 years, approximately 565 h of frontal-view recordings of individual signers have been collected. What is important is that these elicited recordings were obtained from 150 Deaf PJM signers from all over the country, and the group of informants included people of various ages, places of origin, and gender in equal proportions. On the one hand, the corpus thus developed provides an invaluable source of foundational data for use in ongoing research into the recognition of facial expressions during the broadcast of a message in Polish Sign Language. On the other hand, the development of automatic systems for recognizing faces in footage will enable linguistic research in new areas and will greatly speed up the search for and selection of non-manual data.

All research efforts related to the development of tools for the recognition of sign language (both sign language gestures and non-manual components) and the proper reproduction of this language in the form of images in motion aim to develop fully-fledged automatic sign language technologies and to enable free communication between hearing and Deaf people.

Digital solutions for sign communication and translation are being developed for the fields of medicine [25], security and transportation [26,27] and public administration [28]. Many efforts have also been made in the field of education. Among others, a Turkish project was described in [29], in which the benefits of using a 3D Avatar in the educational process of Deaf children were presented. For the purpose of the experiment, an avatar was created and a test was performed using it to compare the educational effectiveness of the avatar with text-based educational tools. The results indicate that avatar-based tutoring was more effective in assessing the child’s knowledge of certain words in a sign language. 3D avatars are also being used to teach specific electrical engineering concepts in Portuguese Sign Language (LIBRAS) [30], to present content from a Mexican history textbook for elementary 4th grade in Mexican Sign Language [31] or to create digital math educational materials in American Sign Language and Arabic Sign Language [32,33]. Thanks to automated tools, it is also possible to learn the basics of sign language (ASL) on one’s own. A computer system has been developed in which, based on a neural network, it is possible to classify fingerspelling alphabet letters recorded with a webcam [34].

In the area of education, one can also consider the potential of using different types of dictionaries such as Arabic [35] or Indian [36] and automatic translators such as Paula, the Avatar of English Sign Language, being developed at DePaul University [37].

The technological solutions currently under development must address the problem comprehensively, considering the current state of knowledge about Deaf communities and culture, their needs and experiences, and actively including Deaf sign language users in the research. It is also necessary to be aware of the complexity of the construction and functioning of sign languages and to consider this fact in developing related technological tools. A review of the presented publications showed that researchers are aware of the existing problems and are making every effort to deal with them.

The presented solution can be applied, among other things, in linguistics (corpus research, annotation, construction of sign language dictionaries), didactics (teaching of mimic expression during sign language communication and verification of its correctness), comparative social research (search and comparison, mimic characteristics in given sign communities), or creation of technical solutions for sign language visualization.

Therefore, in this work, we propose the use of action units as support units for the automatic annotation of facial expressions in the sign language translation process. The main aim of this work is to evaluate entropy in real translation recordings and analyze the data associated with the detected action units. This is an important contribution because it is not possible to annotate every frame of the recording. Our work, among other things, is an analytical contribution, allowing us to check the loss of information in real recordings of sign language signers labelled with the use of action units. This paper is also concerned with the analysis of the relationship between the different action units in real data sets—recordings of a sign language signer. We also present the whole process carried out from recording to the annotation of action units. Our discussions are supported by the expertise of Polish Sign Language experts and their indications of the possible application of the detected action units in a larger automatic sign language translation project.

The article is organized as follows: In the next section, we present the theoretical background of the research. The literature review is discussed as well. The third section presents our research methodology. The fourth section includes the numerical experiments and the generated results. The discussion is included after the numerical experiments. The two last sections include a short summary and details of future research.

## 2. Background

The research described in the paper has been undertaken within the Avatar2PJM (Project: Framework of an automatic translator into the Polish Sign Language using the avatar mechanism, The National Centre for Research and Development, GOSPOSTRATEG-IV/0002/2020) project. The goal of the project is to develop a framework that would allow to translate of utterances in the Polish language to Polish Sign Language using an avatar and artificial intelligence methods. The innovative character of the solution lies in consideration of emotions and non-verbal elements of the utterance in the visualization of gestures. The basis for launching the research is the need to develop a solution that will increase the activity of the deaf and contribute to the liquidation of social barriers that these people face. These problems can be overcome by providing the deaf with the tool to support communication in their native language (Polish Sign Language, PSL). The project has been commissioned by the Chancellery of the Prime Minister and will be carried out by the Łukasiewicz Research Centre—Institute of Innovative Technologies EMAG and the Institute of the Polish Language of the Polish Academy of Sciences.

The project assumes the development of a method to translate the Polish language into Polish Sign Language along with a mechanism to control the avatar application. The sign language avatar is a computer representation (animation) of language phenomena. Thanks to good video-recorded reference material, it is possible to animate any described utterance. That is why one of the research stages dealt with acquiring the largest possible number of video recordings of a sign language translator within a Motion Capture (MoCap) session. MoCap is a technique of recording the three-dimensional movements of an actor. It is used in computer games and imitates the natural movements of objects or people in a very realistic way in order to achieve a natural effect. In the case of sign language avatars, MoCap allows copying the signs of the sign language and increasing the comprehension of the communicated content because, from the animator’s perspective, the uttered signs of the sign language consist of geometrical poses and movements.

A sign language message consists of sign language signs and different additional information because what is expressed physically results from coexisting linguistic and non-linguistic processes. While producing computer-generated animations, the emotional context of the utterance is taken into account, along with such phenomena as proper lips movements or voiceless speech, which are performed during sign language messages. What is particularly important here is the sign language interpreter’s facial expressions and the information they convey. Such elements are also significant in the context of data indispensable to developing the translation module. The material acquired during the MoCap session is used to feed the animation module and acquire a set of input data for the translation module based on machine learning methods. To make it possible, it is necessary to submit the set of recordings to the annotation process. Annotation describes particular elements of sing-language signs in particular time intervals of the sign duration. In addition, the process describes singular signs, dictionary-based interpretations (lemmas, lexemes), and information, e.g., about non-manual elements of a sign. Because one of the key annotated elements is the sign language interpreter’s face, and the process is very time-consuming, the researchers attempted to examine the possibility of automatic recognition of the translator’s mimic poses. Such automatic annotation would significantly improve and speed up the annotator’s work. This paper describes the partial results of research undertaken in this domain.

One of the expected effects of the project is pilot testing in selected on-line information services run by the administration. A common use of automatic translation mechanisms in public-service internet systems will be a constructive step to improve the digital availability of public administration. In addition, the project team will examine the career potential of the deaf as well as their satisfaction with contacting public administration before and after the application of the virtual translator. This will allow determining social and economic barriers faced by the deaf while contacting the administration and moving on the job market. The career potential of the deaf will be analyzed and data acquisition methods will be determined to achieve the highest possible professional activation result. The results of the project will allow to permanently liquidate barriers encountered by Polish Sign Language users.

### 2.1. Polish Sign Language—The Role of a Facial Expression

Polish Sign Language (pl. polski język migowy, PJM), like other sign languages, is an autonomous, standard, and fully-fledged natural language, which means it constitutes a two-class system of conventional characters for universal communication. The physical nature of a sign language text, which makes it different from phonic languages, is not vocal-auditory but visual-spatial [38].

An utterance in PJM is composed of manual and non-manual signals. What is essential, the former one play both expressive and linguistic functions in natural sign languages. The expressive function consists of presenting, by articulators such as eyes, eyebrows, mouth, head, and shoulders, various emotional states while performing certain signs (e.g., expressing sadness while showing the SAD sign). However, more important is the linguistic function of non-manual signals, which proves the autonomy of a sign language and allows to distinguish PJM from the phonic Polish language [39]. The linguistic function of non-manual components in PJM consists of several aspects: (1) their phonological construction within signed words, (2) their lexical functioning as independent non-manual signs, (3) their grammatical functioning at the morphological level, and (4) their syntactic functioning in distinguishing signed sentences [40,41]. In the next sections of this chapter, these four aspects of the linguistic function of non-manual elements in PJM will be presented.

#### 2.1.1. Non-Manual Component within Sign Words

At the phonological level, non-manual signals may be obligatorily embedded as an additional element in some signs. It is important to note that the sign of sign language consists of three sublexical parameters that are the sign equivalents of phonemes (cheremes). They are hand configuration, the location of sign articulation, and the movement performed during articulation [42]. Robert Battison [43] added two more properties: palm orientation and non-manual elements to this classification.

Considering the above-mentioned elements, three types of signs are distinguished: manual, multi-modal, and non-manual. Manual signs have only three basic parameters: hand configuration, location, and movement. Apart from these three parameters, multi-modal signs contain the fourth one—a non-manual signal. Non-manual signs are articulated using only non-manual signals without the use of hands and, therefore, can be self-realized at the lexical level and will be discussed in the next subsection [44].

Non-manual features in sign languages include facial actions and expressions, head movements and positions, shoulders, and the position of an upper body as a whole. The area of the body below the waist (hips, legs, and feet) very rarely serves as an active articulator. In our research, attention will be focused on the aspects of a facial expression which involve the eyebrows, eyelids, eye gaze, cheeks, nose, lips, and jaw. These parts of the face can assume the role of independent articulators or can be used simultaneously with the head, shoulders, and the whole body or with manual components [45,46]. Each of the indicated parts of the face can make appropriate movements, which, for Polish Sign Language, were included in the classification of Piotr Tomaszewski, presented below in the form of a diagram (Figure 1) (In our research and further in this paper, we concentrate only on facial expression, and we overlooked other “places” such as head, shoulders, and body—torso—movements.) In it, no other parts of the face, such as the forehead, are marked because they are redundant. For example, due to the anatomical structure, raising the eyebrow always produces a frown of the forehead effect so this latter part of the face is predictable. Each part of the face described as a category of “a place” feature has a set of ways of expressing opinions, feelings, and meanings, defined as “a setting” feature. The most numerous group of settings is boasted of by the mouths, which, both in PJM and in other sign languages, also play an articulatory function with the use of certain signs [41].

Therefore, various mouth actions and their combinations are classified into at least two clearly identifiable types of mouth patterns. These are “mouthings”, which are said to be derived from the surrounding spoken language, and “mouth gestures”, formed within a sign language and, thus, inherent to it [47,48]. Furthermore, different mouth configurations form the basis to create “minimal pairs”, which means that these kinds of signals can distinguish different sign words, e.g., pairs of words such as (BIEGLE “fluent” i.e., in a sentence “Ja migam biegle”) and (SZKODA “too bad” i.e., in a sentence “Nie udało mi się wygrać, szkoda”) are distinguished by different mouth configurations: round vs. stretched [40].

#### 2.1.2. Independent Non-Manual Sign Words

Non-manual signals, e.g., independent non-manual signs, can be used at the lexical level. They refer to signs which do not require the use of the hands at all but employ, in the articulation, only non-manual elements such as specific facial expressions or head movements. An example of a sign using only a facial expression is a sign (NMS: ZGADZA_SIE “That’s right…”), which is articulated by wrinkling the nose, and the examples of signs using only head movements are signs (TAK “yes”) and (NIE “no”) [40].

#### 2.1.3. Role of Non-Manual Signals in Modifying Some Sign Words

Non-manual signals can also be superimposed on a single sign word or a sequence of words, fulfilling a grammatical function, e.g., to form comparative or superlative adjective forms. Non-manual factors may constitute a meaning-modifying (enriching) feature. In order to shorten the statement, the procedure of a simultaneous use of a feature is used, during which, instead of the sequence of characters (“nominal” + “attributive”), only the sign of the object with a non-manually assigned feature is used (e.g., sign DZIEWCZYNA “girl”, NMS: a neutral facial expression, and sign ŁADNA_DZIEWCZYNA “pretty girl”, NMS: smiling face, eyes squinting). A facial expression also allows to express the intensity of a feature. It is a way of intensifying or weakening the meaning of adjectives (e.g., distinguishes signs ŁADNY “pretty”, NMS: calm, smiling face, and BARDZO_ŁADNY “very pretty”, NMS: smiling face, squinting eyes, a slight head movement to the left), and it is also a way of modifying the meanings of verbs (e.g., CHCIEĆ “want”, NMS: a slight nod, lips tightened, and BARDZO_CHCIEĆ “want very much”, NMS: firmly nod, lips tighten, looking up). The handshape of these lexemes remains the same, the only determinant of intensity is the non-manual component [46,49].

#### 2.1.4. Syntactic Functions of Non-Manual Signals in Distinguishing Signed Sentences

Non-manual signals in the syntactic function are essential, especially when distinguishing sentences. Due to the possibility of using non-manual grammatical components, different kinds of sentences with different clauses are formed. Below are a few kinds of sentences and examples of their non-manual components.

-yes/no questions—PJM they can be answered simply by confirming or denying the entire sentence. They are marked in sign language by a slight forward tilt of the head and raising the eyebrows during the whole sentence. That is the only non-manual form that distinguishes the corresponding statement from the question.-“wh” questions—PJM the group of question words used for this purpose includes who, what, where, when, why, what kind of, how many and which. In this kind of utterances, the grammatical non-manual component consists of lowered eyebrows and squinted eyes that occur either over the entire wh-question or solely over a wh-phrase that has moved to a sentence-final position.-negative sentences—a kind of sentence into which signs indicating negation are added. As a non-manual component, there is a relatively slow side-to-side head shake that co-occurs with a manual sign of negation, and the eyes may squint or close.-conditional sentences—they contain subordinate sentences that express the conditions of implementing proposals included in superordinate clauses. In sign languages, subordinate sentences are formed by raised eyebrows, wide eyes, head forward (or back) and tilted to the side, followed by a pause after which the eyebrows and head return to neutral position [39,40,50].

A facial expression, which is a grammatical exponent, imitates the natural facial expressions accompanying the formation of the aforementioned types of sentences. It is also possible to create sign sentences that combine some of the above sentence types [46,49].

### 2.2. Action Units

Action units (AUs) define facial muscle activity so that it is possible to indicate activity that affects facial expressions (facial appearance). The origins of action units are related to the facial coding system proposed in 1978 in the work [51]. In this system, all visually perceivable facial expressions are described. The mentioned expressions are divided according to muscle movements in the following steps.

In the following years, attempts were made to detect units of action automatically, but mainly these were approaches related to a specific expression (happiness, sadness, etc.). However, at the beginning of the 19th century, with the increase in computational capabilities and thus the development of machine learning algorithms and computer vision, work was taken on more complex AU [52].

With the development of more machine learning methods, including deep learning, automatic AU detection became more and more precise [53,54,55]. This makes it possible to apply models learned to detect AU to real-world problems (including real-time detection). In general, however, many of the tools only detect a limited number of AUs. In this way, the responsible action units shown in Figure 2 are most often recognised. In this case, we have additionally subdivided the detected action units due to aspects related to sign language (this is a subdivision for the execution of a movement with the corresponding part of the face).

In this work, we aim to analyze the application of AU to the real-world problem of automatic facial expression annotation in the sign language translation process. Therefore, we focus on using existing algorithms for AU detection [57] and rely only on the AU detected by this tool, as described above.

## 3. Research Methodology

This article aims to analyze action units in the context of their use in automatic text-to-sign language translation (using a specially prepared avatar that signs appropriate sequences in sign language and performs body movements and facial expressions). Our approach is one of avatar control and concerns facial expressions, so we focus only on the face of the signer.

For the application of action units (AUs), analyses related to the entropy of specific AUs depending on the recording, as well as correlation, were applied. The aim is to create rules to correlate specific AU sequences with facial expression elements important for sign language annotation.

In this work, we analyze the real recordings related to the Avatar2PJM project described in Section 2. Consequently, our work is also related to image processing. Figure 3a shows one frame of the actual recording. In our case, we only consider facial expressions, so the identification of the face itself and the removal of the background must be made (Figure 3b). Only the image prepared in this way is used to find action units.

The next step of the analysis carried out is AU detection. At this stage of the work, we are using an off-the-shelf and tested tool for researchers working on computer vision and machine learning for AU analysis—OpenFace 2.0 [57]. The approach used gives better accuracy for detecting face landmarks and face action units than the OpenFace [58] tool. The method is based on linear support vector machine learning. However, it has been shown in the work [57] that the results obtained are similar to methods based on deep learning. It is worth noting that, in the presented solution, AU is detected in two ways. The first one (called presence) is about detecting whether AU is found in a given frame of the video (0—not found, 1—found), while the second one (called intensity) determines the intensity of occurrence of a given AU (from 0.0 to 1.0, where the closer to 0.0, the lower the intensity and vice versa). It should be noted that both models were trained on different learning datasets (described in more detail in [57]), so different results are possible.

The use of OpenFace 2.0 allows for real-time analysis – which is essential for our project. Additionally, it is based on approaches [59,60,61], but for our work, the most important issue is the quality of AU detection. In this case, the authors in the [57] paper showed that the solution we used shows better results (reported as Pearson correlation coefficient) than other popular solutions based on, among others, the convolutional neural network.

The result of this part of the work was to obtain a table consisting of columns describing each frame of the recording with 40 features related to, among others: frame number, time stamp, and AUs. Depending on the method, these are different AUs for presence: 1, 2, 4, 5, 6, 7, 9, 10, 12, 14, 15, 17, 20, 23, 25, 26, 28, and 45; and for intensity: 1, 2, 4, 5, 6, 7, 9, 10, 12, 14, 15, 17, 20, 23, 25, 26 and 45. As our work is related to the analysis of the applicability of AUs in automatic sign language translation, the AU results have been grouped, as suggested by Polish Sign Language practitioners, due to the part of the face. They are matched with the corresponding recording frame, as shown in Figure 4.

It should be noted that the recordings used in our analyses consisted of 25 frames per 1 s. Labeling facial expressions consistent with Polish Sign Language proved to be impossible based on a single frame of the recording. Therefore, in the next step, it was proposed to prepare average AU values for one second of recording—this gave 25 frames in one comparison. The results of this approach are presented in Figure 5.

The Polish Sign Language experts showed that the facial expression changes too often for one second of recording. This is also evident in Figure 5. Therefore, analyses were also carried out for a smaller number of frame of the recordings. As a result, finally, the average AU values for 5 consecutive recording frames were determined. The final result of the data preparation is presented in Figure 6.

## 4. Results

The experiments aimed to test the relationship between AUs in real sign language recordings. In order to do this, recordings made for the Avatar2PJM project were analyzed—a total of several hundred megabytes of recordings (25 frames per second). During our analysis, a number of possibilities were tested—we also made comparisons between the recordings of individual utterances in relation to the material as a whole. This was all done to enable us to find correlations between AU and Polish Sign Language signs in the future—during annotations for automatic translation.

All recordings were processed according to the steps described in Section 3. Consequently, tables were created for each set of recordings describing:each frame of the recording;the average of 5 consecutive frames of the recording (with the frames averaged offset, so first frames [1, 2, 3, 4 and 5] then [2, 3, 4, 5 and 6] etc. up to [n−4, n−3, n−2, n−1 and *n*], where *n* is the number of all frames in the recording;the average of 25 consecutive frames in the recording (analogous to the 5-frame approach).

For the results presented in this work, tables were created containing respectively: 20,211, 20,127 and 19,665 rows. This is the result of combining several different recordings, hence offsets reduce the number of frames (and thus indirectly the rows in the table). In each row, AU-related labels have been extracted—depending on the method chosen (presence or intensity).

### 4.1. Degree of Variability of the Data

As a result of our analyses, we used entropy (see Equation (Equation 1), where we are dealing only with binary AU values, where *p* is the probability of one of these values occurring—the probability that an AU occurs or does not occur in the recording) as a measure of variability in the data. To do this, we determined the entropy value for each of the designated AUs over all the recordings. In this way, we are able to decide on how much the value of each AU changes—an alternative to this is the histogram. Still, by using entropy, we can indicate a gain or loss in variability, depending on our approach (single frame, 5 frames and 25 frames analysis).
(1)$$E\left(p\right)=-p\log_{2}p-\left(1-p\right)\log_{2}\left(1-p\right).$$

It should be noted that for the present approach, determining the entropy value was not a problem—there are only two values of each feature (of each AU) in these data. In the case of the intensity approach, on the other hand, we proposed to normalize the data so that all values less than or equal to 0.5 were labeled, 0, and greater than 0.5 were labeled 1. Ultimately, in both cases, there are two possible values of the feature so that when the entropy is 0.0, the data are maximally ordered (there is only one AU value). When the entropy is 1.0, the data are maximally unordered (i.e., an AU is found as many times as it is not found).

Table 1 and Table 2 record the entropy values for each AU according to analyzing each frame separately, averaging over 5 frames and averaging over 25 frames (analogous to the description in Section 3). As can be seen, for the present approach, there are essentially no differences in the entropy of each AU between the situation when each frame is analyzed separately and five frames in sequence. Thus, it can be said that averaging the AUs with an offset of five frames, i.e., $\frac{1}{5}$ s, allows for a better analysis of the prepared data (see Figure 6) by the expert while maintaining an identical distribution of information.

The situation is slightly different when averaging 25 frames, i.e., analyzing the entire second. In this case, for as many as 10 AUs out of 18, there is a change of more than 0.04, with a change of more than 0.1 once (in the case of $AU_{12}$, the lip corner puller). In only two cases is the change close to 0 (this is the case for $AU_{4}$ and $AU_{5}$, i.e., information related to the elevation or lowering of the eyebrows). This indicates a change in the information in the data for the vast majority of AUs.

Similar correlations are found for the intensity approach, although in this case differences also appear when changing from 1 frame to 5 frames. This has to do with normalization; even so, the difference does not exceed 0.025 (in the case of $AU_{5}$, so raising the eyelids is precisely 0.0241). In total, only in 5 cases does the difference exceed 0.01.

Similarly, when comparing 1 frame with 25 frames, the differences are higher than for the present approach. In this case, only twice are they smaller than 0.02 (not once are they smaller than 0.01). In contrast, in as many as 8 cases (out of 17) the difference exceeds 0.1, where for $AU_{5}$ it is more than 0.18, and for $AU_{17}$ (chin raise) it exceeds 0.21.

Our research indicates that it is possible to use the approach shown in Figure 6 to support the work of the expert when investigating AU mapping rules in facial expressions used in Polish Sign Language. Therefore, in the following steps, it is possible to focus on the approach related to the analysis of five frames of recordings. In this case (and the present approach), it can be observed that for some AUs the entropy value is very high (above 0.8 and often close to 1.0), but in some cases, the entropy is low. For example, for $AU_{28}$ (lip suction), the entropy is only 0.0669, so the repeatability is very high. It is also possible to distinguish $AU_{6}$, $AU_{7}$ and $AU_{9}$, i.e., action units related to the cheeks and nose.

Interestingly, the intensity approach is characterized by slightly different values—this has to do with the normalization of the values but also with the different ways of training for each approach. Here, the lowest of the entropies relates to $AU_{5}$ and is 0.3 (in the present approach, it was close to 1.0). It can be seen that for the intensity approach, a certain repetition of AU is noticeable in more cases.

### 4.2. Correlation of Action Units

Figure 7 and Figure 8 present correlation heat maps for both approaches using five frames. Based on expert knowledge and experiments from Section 4.1, in this section, we present only the analyses for five frames.

As we can see, there are also differences between the two approaches used in the case of correlation. In the case of the present approach, the correlation between AUs is mainly noticeable between $AU_{25}$ and $AU_{26}$ (mouth opening and jaw lowering—this is a well-known relationship), but the situation is more interesting for the correlation between $AU_{4}$ (eyebrow lowering) and $AU_{45}$ (blink) and $AU_{6}$ (cheek raiser) and $AU_{12}$ (lip corner puller). This indicates that more correlations can be found beyond the classical correlations during sign language recordings, where facial expressions are very significant and often emphasized. It is also possible to notice a considerable lack of correlation between, for example, $AU_{4}$ and $AU_{5}$ or $AU_{17}$ and $AU_{25}$, i.e., opposite facial expressions. This confirms that the use of AU in the sign language translation approach may be more relevant due to the high emphasis on facial expressions by sign language speakers.

In the intensity approach, on the other hand, there is a correlation between AU in a much higher number of cases. This is a signal to experts that when using the intensity approach, AUs are more likely to occur simultaneously. It should be noted that for these analyses, the AU values were not normalized in any way. The highest correlation concerning the other AUs is shown by $AU_{6}$. A significant correlation of $AU_{6}$ is seen with respect to $AU_{7}$, $AU_{9}$, $AU_{10}$, $AU_{12}$ and $AU_{14}$. Similarly, as with the present approach, a significant lack of correlation occurs between $AU_{17}$ and $AU_{25}$.

## 5. Discussion

In our work, we have seen an opportunity to use action units to mark facial expressions during sign language translation. However, to enable the work of experts, it is necessary to adapt the real data, in the form of recordings, for expert analysis. To this end, we proposed a comparison of AUs found from two different approaches with real video recordings. A comparison of the entropy of these data shows that it is possible to use five more frames of footage simultaneously (with averaged AU values). This makes the determination of facial expressions more precise and allows for a deeper analysis of the expert’s knowledge of AU values.

It has also been shown that the correlation between the individual AUs in the case of recordings of sign language signers shows other relationships in addition to the classical ones. This is due to the high intensity of facial expressions and the particular emphasis of facial expressions on individual gestures. Accordingly, the use of AU labeling is justified in the context of future facial expression labeling during automatic sign language annotation.

An effective AU recognition program will be able to be successfully used in linguistics. In particular, in corpus studies of Polish Sign Language. It will make it possible, for example, to search corpus texts for the occurrence of a given facial expression or element of facial expression and to analyze this occurrence in connection with the sign signs at which the expression appears. This solution will also make it possible to build sign language dictionaries (in particular, Polish Sign Language, which is of interest to us), in which content searching can be carried out based on the indicated facial expression element or combinations of indicated features of individual facial parts forming the selected facial expression system (Search-by-video sign language dictionaries).

Using such a solution will benefit linguistic research but will also make it possible to develop and implement new methods to teach the correct reading of mimicry in PJM-signaled communication by deaf people, as well as to teach proper mimicry expression in conjunction with signaled manual communication.

Due to the implementation of the Avatar2PJM project, the team’s further work on developing the tool for automatic translation into Polish Sign Language will focus on using it to automate the annotation of Polish Sign Language in ELAN. The inspiration for the work on automated annotation came from the very involved and time-consuming work of annotating a large corpus of sign language texts needed for the next stages of project work. Therefore, we began analyzing the possibility of automating this process. The planned end result of the work is the possibility of automatically generating records in ELAN with a precise indication of time intervals and annotating predefined non-manuals in them. The basis for showing non-manuals in a given time interval is the achievement of a sufficiently high AU detection rate in a given time interval of a recording with sign language content. Non-manuals annotated in ELAN would be identified by the researcher as a closed catalogue before the automatic annotation process begins as “controlled vocabulary” with respect to the individual assumptions of a given project. An example of a catalogue of non-manual names for the Polish Sign Language is Figure 1, where “place” denotes the annotated part of the signer’s face (annotated in separate ELAN layers due to the potential for overlapping intervals) and “settings” are lists of possible annotated changes in the layout of that facial area.

The research carried out so far shows the potential and scope for using AU in automatic annotation, but the execution of such a tool requires further research and implementation work. The analyses performed so far have been conducted on real-life recordings, but our trials demonstrate that the tool under development will be able to successfully annotate recordings of a very different nature, such as video excerpts, found data from sign corpus, social media materials, etc. These recordings should be prepared in advance and meet the indicated criteria, e.g., regarding the quality/resolution of the recordings. Also important is its ability to be used to annotate different sign languages, as the method of annotation can be determined by the researcher himself and adapted to the non-manuals present in a given sign language. Once developed, the automatic annotation model will therefore be replicable for different sign languages.

In the model presented here, it is necessary to map AU detection in individual facial areas to the corresponding names of non-manuals specified for a given sign language. On the other hand, the tangible result of the implementation of the automatic annotation tool is the ability to quickly acquire a large number of annotated recordings in terms of the non-manuals present in them, which creates a large input database for the development of the automatic translation tool. The final result of the automatic annotation tool of sign language non-manuals is ready-made data files that are input for the automatic translation tool.

We are also aware of some limitations in the use of the described method and the need for further analysis of the possibility of using AU in recognizing facial expressions in Polish Sign Language.

In our study, we used 18 AUs responsible for recognizing muscle activity in different parts of the face. After expert analysis, it can be concluded that this number of AUs is insufficient to mark all facial expressions that are used during communication in Polish Sign Language. Especially complex seems to be the aspect of mouth actions. Hanke [62] points to an extensive list of 59 “mouth gestures” recognized in research on British, Dutch, and German Sign Language. This should be included the large variety of “mouthings” as non-manuals of sign language related to articulation in spoken language. Therefore, it will be necessary to find solutions dedicated specifically to this area of faces. On the other hand, we note that the developed method allows for the marking of crucial facial expressions, thus, providing a chance for at least partial automation of work during the translation or annotation of sign language.

## 6. Conclusions

This work aimed to analyze actual sign language recordings in terms of the use of action units in the automatic translation of the text into Polish Sign Language. This is part of the Avatar2PJM project, in which correct translation into sign language with appropriate facial expressions is an important aspect.

In this work, we analyzed the entropy of action units in real recordings and its change when averaged—which has to do with the labeling of recordings by experts. Marking at $\frac{1}{25}$ s is impossible, but $\frac{1}{5}$ s is already a sufficient time range for experts. It was shown that the entropy does not change significantly when considering five frames of recording, allowing further work on the design.

The correlation between each action unit and the frequency of occurrence of each was also determined. This is valuable information for experts who are working on finding rules to map known action units in facial expressions required for translation into Polish Sign Language.

Future work will develop rules to label facial expressions based on detected action units. In the case of the intensive approach, we will also examine the use of certain margins when determining intensity—known from the three-way decision theory. In addition, it is worth considering the approach of using machine learning directly to find the repetition of the labeling (done by the experts) without using action units. For this purpose, however, a sufficiently large dataset should be prepared.

## Figures and Tables

**Figure 1 entropy-25-00120-f001:**
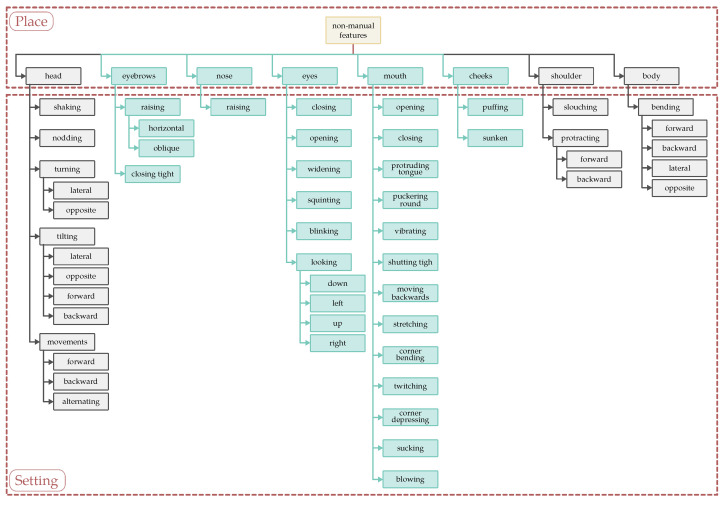
Place and settings features of non-manual components in PJM signs.

**Figure 2 entropy-25-00120-f002:**
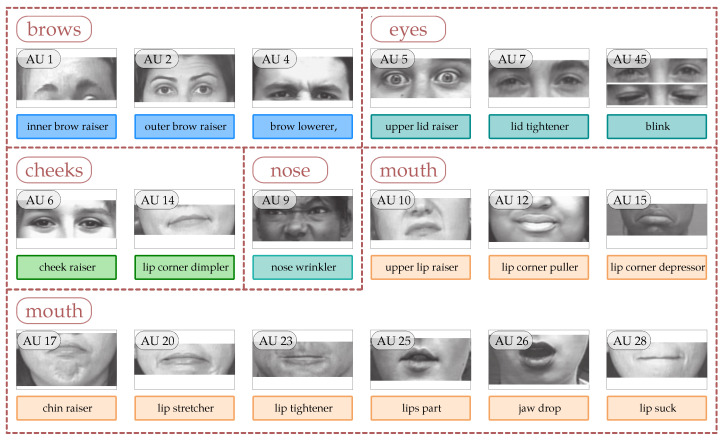
Action units with sign language-compatible facial parts—images from [56].

**Figure 3 entropy-25-00120-f003:**
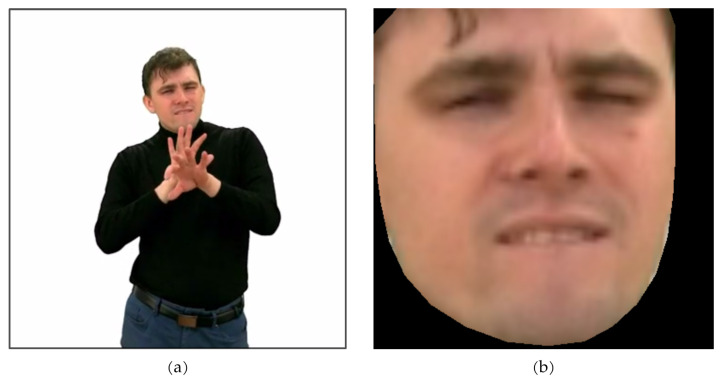
The real data used in this work: (**a**) Image before face extraction. (**b**) Image after face extraction and background removal.

**Figure 4 entropy-25-00120-f004:**
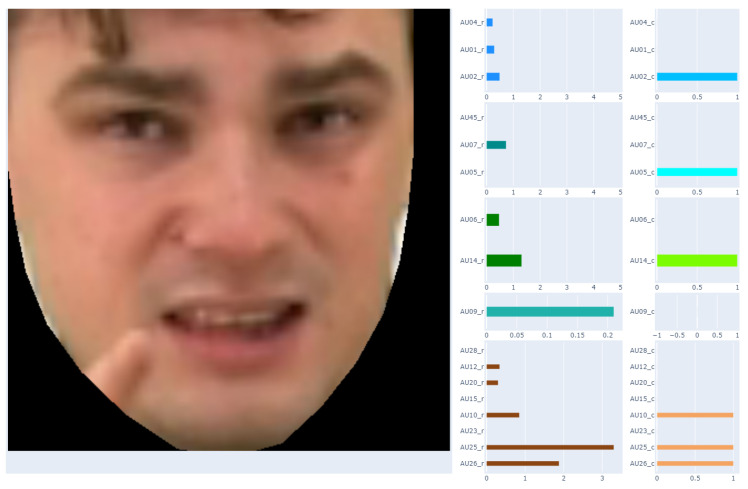
Matching AU with the corresponding recording frame.

**Figure 5 entropy-25-00120-f005:**
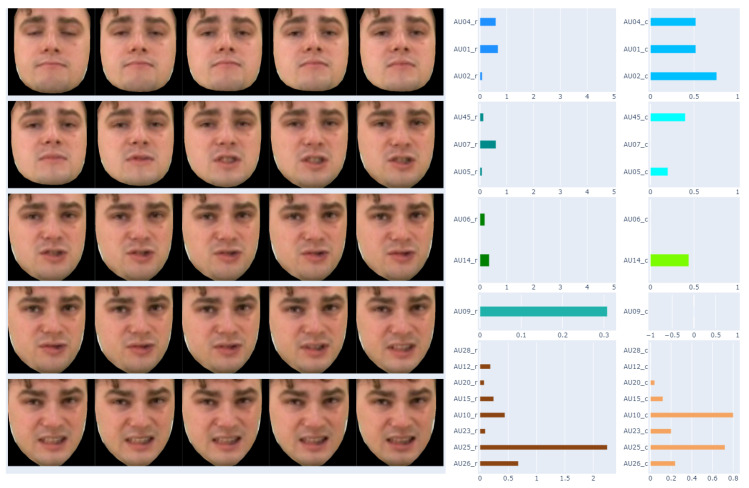
Matching AU with the corresponding one-second (5 frames) of the recording.

**Figure 6 entropy-25-00120-f006:**
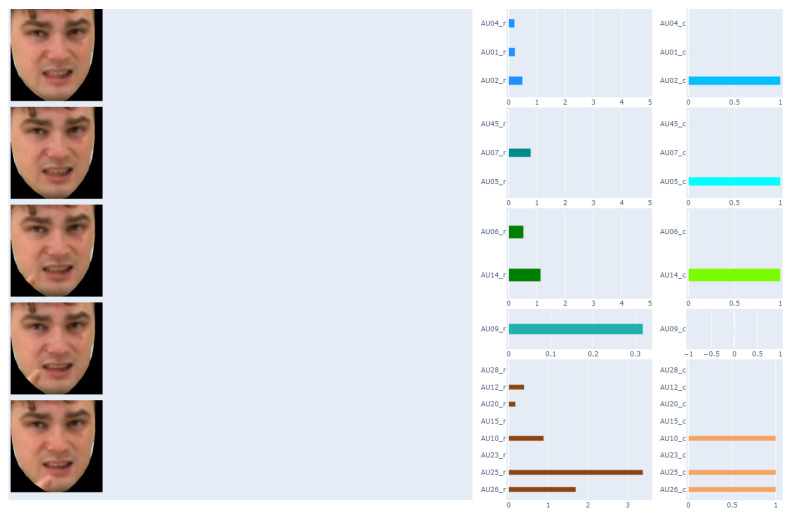
Matching AU with the corresponding 5 frames of the recording.

**Figure 7 entropy-25-00120-f007:**
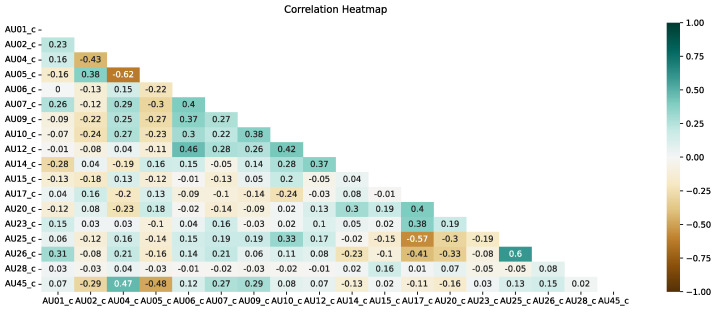
Correlation heat map for the presence approach and the use of an average of 5 frames.

**Figure 8 entropy-25-00120-f008:**
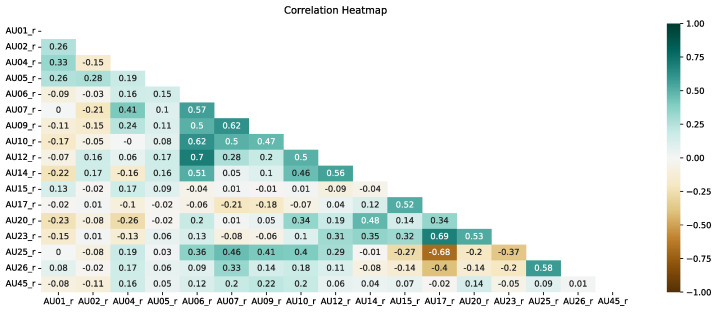
Correlation heat map for the intensity approach and the use of an average of 5 frames.

**Table 1 entropy-25-00120-t001:** Entropy value for each of the action units for the present approach.

Action Unit	1 Frame	5 Frames	25 Frames
$AU_{1}$	0.7351	0.7354	0.6723
$AU_{2}$	0.9407	0.9398	0.9231
$AU_{4}$	0.9996	0.9997	0.9999
$AU_{5}$	0.9984	0.9985	0.9990
$AU_{6}$	0.2660	0.2659	0.1987
$AU_{7}$	0.3489	0.3469	0.2768
$AU_{9}$	0.4081	0.4089	0.3309
$AU_{10}$	0.9666	0.9670	0.9617
$AU_{12}$	0.5575	0.5561	0.4554
$AU_{14}$	0.8885	0.8876	0.8420
$AU_{15}$	0.9057	0.9063	0.8772
$AU_{17}$	0.9659	0.9641	0.9338
$AU_{20}$	0.8956	0.8932	0.8329
$AU_{23}$	0.8357	0.8328	0.7647
$AU_{25}$	0.9172	0.9183	0.8900
$AU_{26}$	0.8238	0.8252	0.7472
$AU_{28}$	0.0684	0.0669	0.0265
$AU_{45}$	0.7198	0.7206	0.6368

**Table 2 entropy-25-00120-t002:** Entropy value for each of the action units for the intensity approach.

Action Unit	1 Frame	5 Frames	25 Frames
$AU_{1}$	0.7721	0.7649	0.7506
$AU_{2}$	0.3691	0.3581	0.3152
$AU_{4}$	0.9768	0.9765	0.9873
$AU_{5}$	0.3236	0.2995	0.1399
$AU_{6}$	0.4370	0.4264	0.3061
$AU_{7}$	0.9050	0.9073	0.9274
$AU_{9}$	0.4542	0.4327	0.3109
$AU_{10}$	0.8671	0.8671	0.8927
$AU_{12}$	0.5216	0.5001	0.4184
$AU_{14}$	0.6244	0.6150	0.5585
$AU_{15}$	0.6726	0.6562	0.6267
$AU_{17}$	0.9199	0.9023	0.7031
$AU_{20}$	0.6407	0.6231	0.5620
$AU_{23}$	0.5439	0.5208	0.4196
$AU_{25}$	0.9904	0.9833	0.8872
$AU_{26}$	0.9957	0.9983	0.9777
$AU_{45}$	0.6477	0.6419	0.5226

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
