# Peer review of "Processing Real-Life Recordings of Facial Expressions of Polish Sign Language Using Action Units"

_entropy, 2023, doi:10.3390/e25010120_

Round 1
Reviewer 1 Report
The topic of the paper is very interesting and extremely important. The paper deals with the problem of including facial expressions in the automation of Polish-to-Polish Sign Language (PJM) translation. The presented approach explores the possibility of using action unit (AU) recognition in the automatic annotation of recordings.
The paper contains six sections. The structure of the paper is correct, and the literature review is very broad and comprehensive. All aspects considered are presented in a clear, correct and interesting manner.
My minor comments concern:
- In the introduction, please emphasize more strongly the main contribution of the work.
- It would be good to indicate the use of words in Polish in the situations like "[BIEGLE "fluent"], [HAPPY "too bad"]" and similar cases as well.
- On page 10 please change “p2, 3, 4, 5 and 6]” -> “[2, 3, 4, 5 and 6]”
- Perhaps in future work, a more rough approach to determining the value of an attribute could be tried. I mean, "In the case of the intensity approach, on the other hand, we proposed to normalize the data so that all values less than or equal to 0.5 were labeled, 0, and greater than 0.5 were labeled 1." It is possible to leave some margin (0.5-p, 0.5+p) of uncertainty and treat values within that range differently. This is a standard approach from the three-way decision theory.
Author Response
All responses for remarks are attached as a file.

Reviewer 2 Report
The paper presents a method that processes the real-life recordings of Polish sign language. The authors try to leverage manual gestures and facial expressions to automatically translate the Polish sign language. After carefully reviewing this work, some concerns have been raised as follows:
1. In Sec. 2.2, the authors mentioned 18 AUs that are most often recognized. The reviewer suggests using a table that uses a photo to illustrate these descriptions for each AU.
2. The matching method to identify which AUs were detected in the real-time recordings may be problematic. The details about the matching algorithm are missing. The performance of various matching methods is not compared or discussed. Which performance metric is preferred for AU detection?
3. In the context, it seems no manual gestures are discussed. The title of this work may mislead the readers.
4. The limitations of this work should be clearly pointed out.
5. In Eq. (1), the parameter p is not defined.
6. If the planned future work, i.e., automatic translation into Polish Sign Language, could be presented in this paper, the value of this work would significantly increase.
Author Response

(The authors gave the same response as above.)

Reviewer 3 Report
The manuscript studies the viability of using action units for sign language applications. The authors use an open-source AU detector and run it on a sign language dataset. They evaluate the entropy and correlation of the AU and the signs using different numbers of frames. The results show that some AUs can be good predictors of sign language recognition.
The work presented forms part of a larger project called Avatar2PJM, it is well-written, and the results are not bad. However, I think the significance of the work is not enough for a Journal publication. I would like to see how the AUs perform as features for an automatic translation system.
The work presented can be categorized as a feature engineering or feature selection approach, which is not even necessary in the case of a deep network where you can feed all the facial features, and the model will learn how to use them efficiently.
Author Response

(The authors gave the same response as above.)

Round 2
Reviewer 1 Report
All my comments have been taken into account. A very good publication. Only one thing I noticed that on pages 2 and 5 the word "non-manual" should be written with a capital letter (after the dot) and now it is in lower case.
Reviewer 2 Report
All my previous concerns have been addressed.